# The experiences of people living with obesity and chronic pain: A Qualitative Evidence Synthesis (QES) protocol

Natasha S. Hinwood[1,2]*, Maire-Brid Casey[3], Catherine Doody[1,2], Catherine Blake[1,2], Brona M. Fullen[1,2], Gráinne O'Donoghue[1], Colin G. Dunlevy[4], Susie Birney[5], Fionnuala Fildes[6], Keith M. Smart[1,2,7]

**1** UCD School of Public Health, Physiotherapy and Sport Science, University College Dublin, Dublin, Ireland, **2** UCD Centre for Translational Pain Research, University College Dublin, Dublin, Ireland, **3** TCD Discipline of Physiotherapy, School of Medicine, Trinity College Dublin, Dublin, Ireland, **4** Centre for Obesity Management, St. Columcille's Hospital, Dublin, Ireland, **5** Irish Coalition for People Living with Obesity (ICPO), Dublin, Ireland, **6** Independent Patient Insight Partner, St. Vincent's Private Hospital, Dublin, Ireland, **7** Physiotherapy Department, St. Vincent's University Hospital, Dublin, Ireland

* natasha.hinwood@ucdconnect.ie

**Data Availability Statement:** No datasets were generated or analysed during the current study. All relevant data from this study will be made available upon study completion.

## Abstract

### Introduction

There is a substantial and progressive association between chronic pain (CP) and living with overweight or obesity. The relationship between obesity and CP is intricate and complex, with obesity being associated with increased pain-related disability, pain intensity, reduction in physical functioning and poorer psychological well-being. A Qualitative Evidence Synthesis (QES) provides an opportunity to better understand and reveal key areas within the patient experience of these complex interactions to inform best practice and future intervention design.

### Aims

The aim of this QES is to methodically and systematically review and synthesise the qualitative literature reporting on the personal experiences of people who are both living with obesity (PwO) and chronic pain.

### Methods

The phenomenon of interest of this QES is the lived experiences of PwO and CP. The following research question was developed using a modified Population, Intervention, Comparison, Outcome and Study type (PICOS) framework: *"What are the lived experiences of people living with obesity and chronic pain?"*. One review author will conduct a systematic search based on keywords and Medical Subject Headings (MeSH) terms for finding relevant articles in five peer-review databases, from inception to the date of searching. Two review authors will independently apply inclusion and exclusion criteria and screen articles in a two-stage process. The methodological quality of included studies will be assessed using the Critical Appraisal Skills Programme (CASP) tool and data will be extracted using a customised

**Funding:** This research is an investigator-initiated study funded by an unrestricted University College Dublin Ad Astra studentship (KMS and NSH). The funders did not and will not have a role in study design, data collection and analysis, decision to publish, or preparation of the manuscript. Funding website: https://www.ucd.ie/adastrafellows/en/.

**Competing interests:** The authors have declared that no competing interests exist.

**Abbreviations:** *BMI*, Body mass index; *CASP*, Critical appraisal skills programme; *CMP*, Chronic Musculoskeletal Pain; *GRADE CERQual*, Grading of Recommendations Assessment, Development and Evaluation Confidence in Evidence from Reviews of Qualitative Research; *MeSH*, Medical Subject Headings; *NAFLD*, Non-Alcoholic Fatty Liver Disease; *PICOS*, Population, Intervention, Comparison, Outcome and Study type; *PPI*, Public Patient Involvement.

template. We will undertake a thematic synthesis of qualitative data from included studies and report our findings narratively. Confidence in the findings will be assessed based on the Grading of Recommendations Assessment, Development and Evaluation Confidence in Evidence from Reviews of Qualitative Research (GRADE-CER-Qual) approach.

## Findings and dissemination

This study will follow the Preferred Reporting Items for Systematic Review and Meta-Analysis Protocols (PRISMA) and Enhancing Transparency in Reporting the Synthesis of Qualitative Research (ENTREQ) guidelines. It is anticipated that the findings of the review will facilitate a deep and broad understanding of the complex interactions between CP and obesity and will help inform best practice and future intervention design. Findings will be disseminated through journals that undergo peer review, presentations at conferences, engagement with public and patient advocacy groups, and social media.

## Ethics and dissemination

Ethical approval is not required to conduct this review.

## Trail registration

PROSPERO registration number: CRD42023361391.

## Introduction

Obesity is a preventable, multifactorial chronic disease that impairs health and quality of life, and is associated with an increased risk of morbidity and mortality secondary to many non-communicable diseases (NCDs) [1–4]. The prevalence of people living with overweight and obesity has increased such that it is now considered as a global epidemic [4], affecting almost 60% of adults in the World Health Organisation (WHO) European Region [5]. Its causes are multifactorial and complex, and include genetic [6], behavioural [7], environmental and socio-economic factors [8].

Evidence for an association between increased musculoskeletal pain and obesity has been found in several systematic reviews and cohort studies. However, exact population estimates of the prevalence of pain among PwO are not known. One large scale epidemiological survey (n = 1,062,271) showed that people with obesity reported between 68% to 254% higher rates of pain compared with people with a healthy weight. This association remained significant even after controlling for other pain-related medical conditions, with both this study and others indicating a proportional increase in incidence and severity of chronic pain with increasing BMI in PwO [9–11].

In relation to sites of reported pain in PwO, the most common include low back [12, 13] knee [14], foot [15], and shoulder pain [16, 17]. Additionally, obesity is associated with increased likelihood of multisite pain in the lower limbs [18] as well as headaches, abdominal, pelvic pain, and chronic widespread pain/fibromyalgia [11]. A Swedish obesity registry study reported a pain prevalence (pain in five body locations: neck, back, hip, knee and ankles) of 58% among men and 68% among women [19].

People living with obesity and pain have a higher pain incidence, severity and morbidity [1–4, 9–11, 18]. This population also experiences substantially decreased health-related quality

of life when compared to those living with either condition alone [11, 20]. When compared to those living with a healthy weight or underweight, living with obesity is associated with a higher risk of developing joint pathologies such as osteoarthritis and inflammatory arthritis [11]. The associated increased severity of pain is further associated with poorer outcomes in physical functioning–both in terms of quantity and complexity of movements (such as posture or gait patterns) [21–23].

The mechanisms linking pain and obesity are multifactorial [24, 25] and may include interactions between physiological (e.g. inflammatory mediators associated with adiposity), psychological (e.g. social isolation, stress or depression), socioeconomic (e.g. socioeconomic status or education levels), behavioural (e.g. kinesiophobia), biomechanical (e.g. increased kinetic forces or joint load, or decreased muscle mass in sarcopenic obesity) and genetic mechanisms [26–36]. While the case for treating excess adiposity as a strategy for improving multimorbidity has been outlined [37], the mechanisms linking both CP and obesity are complex. Integrating both weight management strategies and chronic pain management approaches leads to improved pain and disability compared with either intervention alone [35, 38, 39].

While the lived experiences of people living with overweight and obesity has been synthesised, the perspectives of only PwO living with chronic pain and understanding the complex interaction between the two has not previously been explored. A previous QES of 32 studies investigated the lived experience of PwO. This study identified five themes, including the development of obesity; the life-limiting effects of obesity; stigma, judgment, shame, and blame; experiences with obesity treatment; and experiences of specific or minority groups [40]. While a valuable addition to the body of knowledge, the scope of this study was very broad and failed to provide any explicit insights into the experiences of those specifically living with both obesity and chronic musculoskeletal pain, as this was not their focus. Another study has synthesised the perceptions and experiences of adults with overweight / obesity and chronic musculoskeletal pain engaging with weight loss interventions (n = 58 participants) [41]. This review highlighted the importance of healthcare professionals understanding the physical and psychological effects of pain that make healthy behaviour more challenging; and acknowledging and discussing the link between weight and pain with their patients [41]. While Cooper et al. 2017 has also provided great insights into the experiences of people overweight and those living with multimorbidity of obesity and CMP, three of the four studies included are now more than a decade old, with several important advancements in the understanding of both pain and obesity in that time. The disease of obesity and disorder of chronic musculoskeletal pain (CMP) are both complex, and weight loss treatments are only a part of the treatment continuum [42–44]. Understanding the interaction between these illnesses through the lived experiences of this population is fundamental to establishing what is most meaningful to this population and highlighting further knowledge gaps and treatment opportunities.

### Aims

The aim of this qualitative evidence synthesis (QES) is to synthesize the available evidence exploring the broad experiences of people living with obesity and chronic pain in mixed settings.

## Methods

### Design

This study is a Qualitative Evidence Synthesis. Understanding the lived experience of people living with obesity and/or chronic pain is fundamental to usefully guide health and social policy and inform the planning and implementation of treatment strategies [44–48].

## Information sources and search strategy

A search strategy (see S1 Appendix) was developed with the assistance of two 'Patient Insight Partners' and an experienced librarian. The search strategy was also informed by a relevant body of literature and guided by the Cochrane Qualitative and Implementation Methods Group [49].

The search strategy was based on appropriate keywords and MeSH terms for finding relevant articles. The following electronic databases will be searched: MEDLINE, EMBASE, CINAHL, Web of Science and PsycINFO from inception to the date of searching. The reference lists of included studies will be searched in order to identify other potentially relevant studies. Grey literature sources will not be searched as our intention is to undertake a comprehensive review of published, peer-reviewed primary research.

**Eligibility criteria.** Our eligibility criteria are defined by a modified PICOS framework (see Table 1) [50].

*Inclusion*. We will include qualitative studies reporting first person experiences of adults (18 years of age or older) living with obesity and chronic pain. Mixed methods studies where qualitative data is reported and extractable separate from the quantitative data will also be included. We will include studies using any qualitative methodology (e.g. phenomenological or grounded theory studies) and using any appropriate method of data collection (e.g. interviews, focus groups). We will include studies undertaken in any country and any setting published in the English language.

*Exclusion*. Studies involving children (17 years of age or younger), non-first person experiences or studies published in a language other than English will be excluded. Studies not published as full research reports (e.g. conference abstracts or editorials) will also be excluded.

**Screening and study selection process.** One reviewer (NSH) will undertake the searches, download the search results to reference manager software EndnoteX V.9 (Clarivate Analytics, Boston), de-duplicate studies using Endnote and import the remaining studies into the systematic reviewing software, Covidence (Veritas Health Innovation, Melbourne, Australia) (Fig 1). After further de-duplication in Covidence, two review authors (NSH and MBC) will independently screen all titles and abstracts identified by the search strategy for eligibility. If the eligibility of a study is unclear from the title and abstract the full-text article will be assessed. The full texts of all potentially eligible studies will be obtained and further

**Table 1. Selection criteria following modified PICOS framework.**

| Modified PICOS | Inclusion | Exclusion |
|---|---|---|
| Population | i. People living with obesity (BMI > 30)<br>ii. People living with chronic pain (>3 months)<br>iii. Adults (18 years and over) | i. People without experience of obesity (BMI <30)<br>ii. Studies involving children (17 years of age or younger)<br>iii. People without chronic pain |
| Intervention | NA | NA |
| Context | Any country or setting (e.g., primary, secondary, or tertiary care, general population) | Perspective of experts working in research area |
| Outcome | Peoples' lived experience of pain | Experiences and opinions of professionals working with people with obesity |
| Study Type | i. Qualitative, including interviews, focus groups, life histories, ethnographic studies, phenomenological studies, grounded theory studies, historical studies, case studies and thematic analysis<br>ii. Focused on personal experience<br>iii. Original research | i. Studies that employ quantitative methods only |

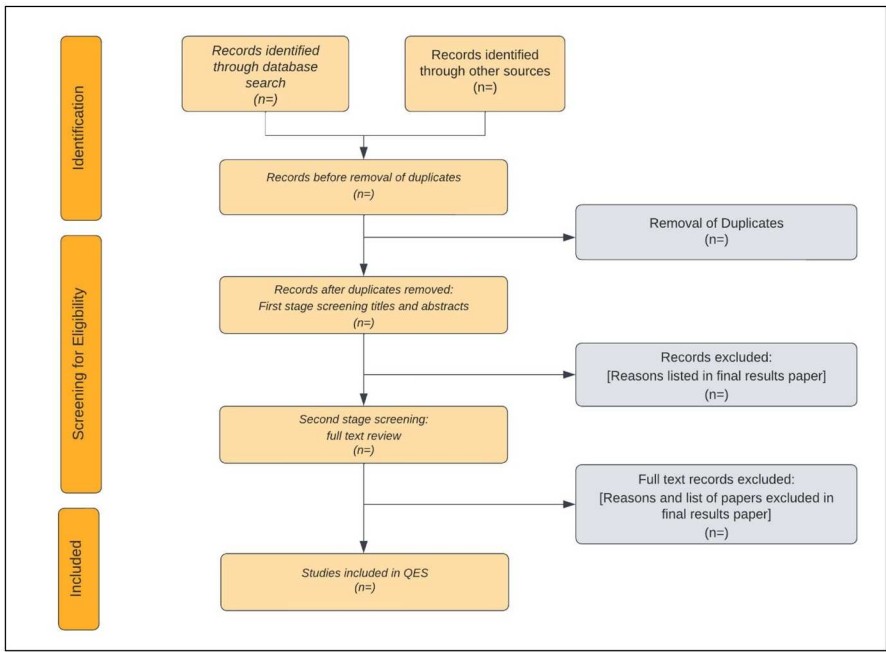

**Fig 1. PRISMA flow diagram for reporting the study selection process.**

independently screened for eligibility by the same two review authors. Any disagreements over which studies to include will be resolved through discussion. If disagreements cannot be resolved, a third review author (KMS) will assess relevant studies and a majority decision made. The search and screening results will be presented in the form of a flow diagram as recommended by the Preferred Reporting Items for Systematic Reviews and Meta-Analyses (PRISMA) checklist [51]. Reasons for excluding full text studies will be reported.

**Data extraction and synthesis.** Two reviewers (NSH and MBC) will independently extract data from all included studies using a customised template. A proforma devised for the review will allow for descriptive data to be captured from the included studies: First author, author characteristics, year of publication, country of origin, study funding, conflicts of interest, ethical approval, study aims, study design, rationale for study and theoretical background, study duration, study setting, sampling and recruitment methods, study inclusion and exclusion criteria, final sample size, method(s) of data collection, use of inductive or deductive coding approaches, data analysis method, themes (and subthemes) identified, and participant characteristics (including age, gender, ethnicity, employment status, BMI characteristics and pain characteristics). For the BMI or weight characteristics, we will aim to extract any information relating to the BMI of included participants from each study, including descriptive statistics (range, mean, standard deviation) where possible. For patient pain characteristics, we will extract information about the severity, duration, aetiology, diagnosis, relevant previous treatment, and location of pain, where available, for participants included in each study.

We will undertake a thematic synthesis in three stages as described by Thomas and Harden, 2008 [45]. In stage one, the original findings of all included studies will be coded, line by line. This will be an iterative and hierarchical process, completed by both reviewers independently initially. The code generation will be completed using Microsoft Excel (Microsoft Excel for Microsoft 365 MSO—Version 2209 Build 16.0.15629.20152). Reviewers will have the option of applying multiple codes for lines of text and will create a mutually agreed taxonomy of codes.

Once all coding has been completed, reviewers will come together to discuss, amalgamate, and compare for interpretive consistency across all coding. Reviewers will add in further categories of coding as necessary. This will ensure that the findings of this review remain grounded in the original studies. A third reviewer will be consulted for resolution of any disagreements. In stage two, the first reviewer (NSH) will draft a summary of the descriptive themes. This will be redrafted based on comments and input from two other reviewers (MBC and KMS) to form a mutually agreed upon summary. While stages one and two more closely align to the findings of the included studies, the purpose of the stage three will be for the generation of analytical themes. This means going beyond the descriptive themes and developing analytical themes which aim to generate new constructs, explanations, or hypotheses [52, 53]. Two reviewers (NSH and MBC) will answer the initial research question posed in the development of the QES for this purpose. Initially this will be done independently, but then mutually together as a group. Through iterative discussion, more abstract themes may emerge. If during the analysis, there are any emergent subgroups (e.g. based on participant characteristics, type of pain, BMI category, or setting), these will be considered for further subgroup analysis. Any identified subgroup data will follow the same synthesis plan as outlined above. Any subgroup analyses will be reported as post hoc decisions.

These themes will then be discussed with the Patient Insight Partners for further input to further assist with contextualization of results and development of meaningful public participation [54]. The results of the synthesis will be used to form the recommendations through consensus between authors.

Thematic synthesis was selected following consideration of the seven domains of the RETREAT framework [55]–(Review Question, Epistemology, Time, Logistic Constraints, Resources, Expertise, Knowledge, Audience and Purpose, and Type of Data) and due to suitability for synthesizing data from different methodologies [55] and aptness for the review purpose [45]. This method has also been used frequently in our field previously [40, 55–57].

**Quality appraisal of included studies.** Two reviewers (NSH and MBC) will independently assess the methodological quality of included studies using the Critical Appraisal Skills Programme (CASP) quality assessment tool for qualitative studies [58]. The CASP tool for qualitative studies has been recommended by the Cochrane Qualitative and Implementation Methods Group and continues to be widely used in similar reviews as it addresses many of the criteria and assumptions underpinning qualitative research [59, 60].

It comprises 10 items with yes/no/can't tell response options that together assess the rigour, credibility, and relevance of qualitative research. Disagreements about study quality will be resolved through discussion. If disagreements cannot be resolved, a third review author (KMS) will assess relevant studies and a majority decision made. Studies will not be excluded based on methodological quality, but quality assessments will be used to determine our confidence in the synthesis findings.

**Data synthesis.** *Assessment of confidence in the review findings.* The Grading of Recommendations Assessment, Development and Evaluation Confidence in Evidence from Reviews of Qualitative Research (GRADE-CER-Qual) approach will be used to assess confidence in each of the findings of this study. The GRADE-CERQual is an approach for assessing the extent to which a review finding is a reasonable representation of the phenomenon of interest and is based on four domains: methodological limitations, relevance, adequacy of data, and coherence [61–64]. Each review finding will be appraised under each of the GRADE-CERQual domains by two reviewers (NSH and MBC).This will initially be done independently, and then both reviewers will discuss together to decide mutually a level of confidence ranging from high, moderate, low, or very low. Following this, confidence in the overall findings will also be assessed and graded as high, moderate, low, or very low [64]. Reviewers will remain sensitive

to possible overlap across components and review iteratively to make a final assessment. Disagreements will be resolved through discussion or if disagreements cannot be resolved, a third review author (KMS) will assess relevant studies and a majority decision made.

In order to enhance transparency, results will be published in a table presenting a summary review of the findings. This table will outline i. each finding; ii. the studies included to support each respective finding; iii. the confidence in each respective finding using GRADE-CERQual as outlined above; and iv. a rationale for the GRADE-CERQual confidence assessment for each finding [65].

*Differences between the protocol and the final review.* Any and all deviations from the QES protocol will be transparently reported.

**Reflexivity.**   *Authors' backgrounds.* The authors are comprised of academic and/or clinical physiotherapists (NSH, MBC, CGD, CD, CB, BMF, GO'D, KMS) with clinical and research interests in pain and obesity. These authors all have varying degrees of experience in conducting qualitative studies or evidence syntheses. Two authors have also been included as 'Patient Insight Partners' (SB and FF), with experience of living with obesity and in advocacy for people living with obesity.

*Bias mitigation exercise.* Prior to commencing the screening and analysis, the two review authors responsible for running the searches, screening for eligibility and then extracting and analysing data (NSH, MBC) will meet to discuss and consider their baseline epistemological viewpoints, preconceptions and beliefs surrounding the subjects of obesity and pain [66]. Additionally, the researchers will undertake a bracketing exercise by discussing what their anticipatory expectations are of the findings. The meeting will be led by NSH and both reviewers will be tasked with considering the relevance of these preconceptions during each stage of analysis. The goal of this reflexive exercise is to reduce or mitigate internally held biases that might otherwise influence decision making or discussion at each stage of the research [67, 68].

**Dissemination of findings and data availability statement.**   We will report this study in accordance with the Enhancing Transparency in Reporting the synthesis of Qualitative research (ENTREQ) statement [69]. We will publish our findings in a peer-reviewed journal, within 12 months of completing the study and further disseminate via relevant clinical and academic conferences, public and patient advocacy groups, and social media. We aim to make our data FAIR, i.e., findable, accessible, interoperable, and reusable [70]. Finally, data generated as a result of the synthesis steps as outlined above, such as generated codes and themes, will be made available as appendices and on the PROSPERO database registration for this study.

## Discussion

The disease burden of pain and obesity has been well-documented. This QES will add new depth to existing knowledge around the complex interaction between obesity and pain [48, 53, 54]. Importantly, our study will give voice to people living with obesity and chronic pain. It is anticipated that our findings will invite healthcare professionals to further consider their assessment and management of pain in people living with obesity and healthcare policy makers, and funders to consider the provision of services and resources to address any potential unmet needs. Addressing the comorbid pain of obesity is essential since people living with obesity tend to respond less well to pain treatments and management compared to people who do not have obesity [11, 39, 71].

## Public patient involvement

Two volunteer 'Patient Insight Partners' have collaborated with and advised the research team in the process of devising our research question and methods; one from the Irish Coalition for

People Living with Obesity (ICPO), and one with experience of attending a Bariatric Clinic at a hospital in Dublin, Ireland. They have provided valuable input on the wider complexities of living with pain and obesity. We are grateful for their continued advice and input on data interpretation, and dissemination of findings. Public Patient Involvement (PPI) will be reported using Guidance for Reporting Involvement of Patients and the Public 2 (GRIPP2) reporting checklist [72].

## Supporting information

**S1 Appendix. Search strategy.**
(PDF)

**S1 Checklist. PRISMA-P 2015 checklist.**
(PDF)

## Author Contributions

**Conceptualization:** Natasha S. Hinwood, Catherine Doody, Brona M. Fullen, Keith M. Smart.

**Data curation:** Natasha S. Hinwood, Maire-Brid Casey, Catherine Blake, Keith M. Smart.

**Formal analysis:** Natasha S. Hinwood, Maire-Brid Casey, Catherine Doody, Catherine Blake, Brona M. Fullen, Gráinne O'Donoghue, Colin G. Dunlevy, Susie Birney, Fionnuala Fildes, Keith M. Smart.

**Funding acquisition:** Catherine Doody, Keith M. Smart.

**Investigation:** Natasha S. Hinwood, Maire-Brid Casey, Catherine Doody, Brona M. Fullen, Gráinne O'Donoghue, Keith M. Smart.

**Methodology:** Natasha S. Hinwood, Maire-Brid Casey, Catherine Doody, Catherine Blake, Brona M. Fullen, Gráinne O'Donoghue, Colin G. Dunlevy, Keith M. Smart.

**Project administration:** Natasha S. Hinwood, Maire-Brid Casey.

**Resources:** Natasha S. Hinwood.

**Software:** Natasha S. Hinwood.

**Supervision:** Catherine Doody, Keith M. Smart.

**Validation:** Natasha S. Hinwood, Maire-Brid Casey, Catherine Blake, Brona M. Fullen, Gráinne O'Donoghue, Colin G. Dunlevy, Susie Birney, Fionnuala Fildes, Keith M. Smart.

**Visualization:** Natasha S. Hinwood, Gráinne O'Donoghue, Colin G. Dunlevy, Keith M. Smart.

**Writing – original draft:** Natasha S. Hinwood, Maire-Brid Casey, Keith M. Smart.

**Writing – review & editing:** Natasha S. Hinwood, Maire-Brid Casey, Catherine Doody, Catherine Blake, Brona M. Fullen, Gráinne O'Donoghue, Colin G. Dunlevy, Susie Birney, Fionnuala Fildes, Keith M. Smart.

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
