## [Decision Letter · Decision Letter 0]

11 Jan 2024

PONE-D-23-17779The Experiences of People Living with Obesity and Chronic Pain: A Qualitative Evidence Synthesis (QES) ProtocolPLOS ONE

Dear Dr. Hinwood,

Thank you for submitting your manuscript to PLOS ONE. After careful consideration, we feel that it has merit but does not fully meet PLOS ONE’s publication criteria as it currently stands. Therefore, we invite you to submit a revised version of the manuscript that addresses the points raised during the review process.

The submission has been assessed by two reviewers, and their comments are available below. Please note that PLOS ONE does consider Study Protocols for publication and these submissions are not rejected for lack of results. Can you please respond to the reviewer comments below?

We look forward to receiving your revised manuscript.

Kind regards,

Vanessa Carels

Staff Editor

PLOS ONE

Journal Requirements:

2. Please amend either the abstract on the online submission form (via Edit Submission) or the abstract in the manuscript so that they are identical.

Multidimensional pain profiling in people living with obesity and attending weight management services: a protocol for a longitudinal cohort study - http://dx.doi.org/10.1136/bmjopen-2022-065188

In your revision ensure you cite all your sources (including your own works), and quote or rephrase any duplicated text outside the methods section. Further consideration is dependent on these concerns being addressed.

Reviewers' comments:

Reviewer's Responses to Questions

**Comments to the Author**

1. Does the manuscript provide a valid rationale for the proposed study, with clearly identified and justified research questions?

Reviewer #1: No

Reviewer #2: Yes

2. Is the protocol technically sound and planned in a manner that will lead to a meaningful outcome and allow testing the stated hypotheses?

Reviewer #1: No

Reviewer #2: Yes

3. Is the methodology feasible and described in sufficient detail to allow the work to be replicable?

Reviewer #1: No

Reviewer #2: Yes

4. Have the authors described where all data underlying the findings will be made available when the study is complete?

Reviewer #1: No

Reviewer #2: Yes

5. Is the manuscript presented in an intelligible fashion and written in standard English?

Reviewer #1: No

Reviewer #2: Yes

6. Review Comments to the Author

You may also provide optional suggestions and comments to authors that they might find helpful in planning their study.

Reviewer #1: The paper is simply the description of PRISMA guidelines, and even made incompletely. I may raise several questions to authors:

1) What is the scientific sense to publish such paper?

2) In which manner it increase the knowledge of the readership?

I think that with out results and complete discussion of such paper it is useless to publish, especially if the the protocol has already been registered and described in the PROSPERO registry.

Reviewer #2: 1. Please define abbreviations upon first appearance in the text.

2. Line 129: please list the five sites in question.

3. Line 131: the formulation of the sentence is ambiguous to the reader.

4. Why the authors didn’t include the Cochrane library to the list of databases to be searched?

5. Why the authors won’t include unpublished studies? Omitting these studies may create bias.

6. Please specify the patients’ characteristics that will be extracted. Will it include the site of pain; duration of pain; level of obesity etc., …?

7. Will the authors conduct subgroup analysis to assess heterogeneity of findings? This analysis should be described in detail.

7. PLOS authors have the option to publish the peer review history of their article (what does this mean?). If published, this will include your full peer review and any attached files.

Reviewer #1: No

Reviewer #2: No

---

## [Author Response · Author response to Decision Letter 0]

19 Jan 2024

Dear Reviewer 1 and Reviewer 2,

We are very grateful to you for all your time and attention to our manuscript. Please find attached our responses to the points raised during the review process. 

Kindest regards,

Natasha Hinwood

---

## [Decision Letter · Decision Letter 1]

18 Mar 2024

PONE-D-23-17779R1The Experiences of People Living with Obesity and Chronic Pain: A Qualitative Evidence Synthesis (QES) ProtocolPLOS ONE

Dear Dr. Hinwood,

Thank you for submitting your manuscript to PLOS ONE. After careful consideration, we feel that it has merit but does not fully meet PLOS ONE’s publication criteria as it currently stands. Therefore, we invite you to submit a revised version of the manuscript that addresses the points raised during the review process.

Thank you for submitting a revised version of the manuscript and responding to reviewers’ comments.

The manuscript is well written and describes a well designed protocol. However some amendments are needed to improve its quality.

In addition to addressing reviewer's comments, please consider my suggestions (listed below) to improve the manuscript:

1.            PICO format: Please revise the population exclusion criteria. I believe the authors missed adding “age less than 18” and “absence of pain” as an exclusion criteria?

2.            Please provide the date range that will be searched

3.            Please include a template for PRISMA flow diagram for reporting the study selection process

4.            Please revise the completed PRISMA-P checklist for any changes the line numbers.

To preserve transparency and uphold the integrity of the scientific process, I would like to acknowledge my reviewer role in the first round of review.

We look forward to receiving your revised manuscript.

Kind regards,

Zeina El Ali, Ph.D.

Guest Editor

PLOS ONE

Reviewers' comments:

Reviewer's Responses to Questions

**Comments to the Author**

1. Does the manuscript provide a valid rationale for the proposed study, with clearly identified and justified research questions?

Reviewer #3: Partly

2. Is the protocol technically sound and planned in a manner that will lead to a meaningful outcome and allow testing the stated hypotheses?

Reviewer #3: Yes

3. Is the methodology feasible and described in sufficient detail to allow the work to be replicable?

Reviewer #3: Yes

4. Have the authors described where all data underlying the findings will be made available when the study is complete?

Reviewer #3: No

5. Is the manuscript presented in an intelligible fashion and written in standard English?

Reviewer #3: Yes

6. Review Comments to the Author

You may also provide optional suggestions and comments to authors that they might find helpful in planning their study.

Reviewer #3: Thank you for the opportunity to review the revision of this manuscript. Obesity and chronic pain are difficult when experienced alone, and exacerbated when experienced together. Much research is needed to understand the complex interactions between the two (note: chronic pain can be considered a disease itself). My main concern is focused on the justification for the aim and explanation of the research question. Specific comments below.

Data Availability Statement

1. The journal specifies requirements for making data available. The authors comment that no data is yet available, please see the two statements and examples from the journal:

a) For protocols without pilot or preliminary data, authors are strongly encouraged to state how they plan to share research data from their study when it is completed or published.

b) For study protocols with data management or sharing plans, authors are encouraged to briefly describe in their Data Availability Statement how data they generate will be made accessible when the study is completed. For example: Deidentified research data will be made publicly available when the study is completed and published.

Introduction

2. Lines 121-122 – change “substantially” to “substantial” or “significant”? (I’m left wondering whether it remained significant).

3. Can you please provide citations for the statement on line 132: “People living with obesity and pain have a higher pain incidence, severity and morbidity.”

AIM

1. How are the statements on lines 150-152 and line 167-168, different? The aim should probably use the phrase “Perspectives of PwO living with CP.” Prior to reading the Aim, the Introduction should definitely elaborate on the distinction between “synthesized lived experiences” and “perspectives of….” It is not clear from the Introduction, so the reader arrives at the Aim without a solid understanding of how this study is different from others, and what contribution it would make to the literature. Based on lines 150-152 of the Introduction, the Aim could be strengthened by saying something about how the data will be used to “understand… the complex interaction between the two.

**It is critical that the authors distinguish between the published reviews of the “lived experience” and what their study will provide/how their study is different. That would help to justify why your study is needed. I recommend using the Introduction to help make your case. To elaborate just a bit further, I read on line 198 that you’ll include studies with “first person experiences…” Can you clearly distinguish between your data vs data from already published syntheses of lived experiences?

2. How will the anthropometric data and pain characteristics data be used to inform your aim?

7. PLOS authors have the option to publish the peer review history of their article (what does this mean?). If published, this will include your full peer review and any attached files.

Reviewer #3: No

---

## [Author Response · Author response to Decision Letter 1]

19 Mar 2024

19.03.24

Dear Editorial Team,

We are very grateful to you and the reviewers for all your time and attention to our manuscript. Please find below our responses to the points raised by the two reviewers during the review process. 

Kindest regards,

Natasha Hinwood

Black text italicised = reviewer comments

Blue text = authors response

Blue text italicised = text in protocol

Red text = amendments to protocol text

Editorial Requests:

1. PICO format: Please revise the population exclusion criteria. I believe the authors missed adding “age less than 18” and “absence of pain” as an exclusion criteria?

This has now been added into the PICO table, but was also originally stated in the main body of the text.

2. Please provide the date range that will be searched

As this is a protocol for planned future work, the final search date will be included in the subsequent manuscript detailing the results. For the purposes of this protocol, we have specified there will be no limits to the search date range.

“The following electronic databases will be searched: MEDLINE, EMBASE, CINAHL, Web of Science and PsycINFO from inception to the date of searching.”

3. Please include a template for PRISMA flow diagram for reporting the study selection process

Thank you for the comment. This has now been included in the protocol. 

4. Please revise the completed PRISMA-P checklist for any changes the line numbers.

Thank you, this has now been revised. Please see included documentation.

Reviewer #3: 

“Thank you for the opportunity to review the revision of this manuscript. Obesity and chronic pain are difficult when experienced alone, and exacerbated when experienced together. Much research is needed to understand the complex interactions between the two (note: chronic pain can be considered a disease itself). My main concern is focused on the justification for the aim and explanation of the research question. Specific comments below.” – Reviewer 3

Data Availability Statement

1. The journal specifies requirements for making data available. The authors comment that no data is yet available, please see the two statements and examples from the journal:

a) For protocols without pilot or preliminary data, authors are strongly encouraged to state how they plan to share research data from their study when it is completed or published.

b) For study protocols with data management or sharing plans, authors are encouraged to briefly describe in their Data Availability Statement how data they generate will be made accessible when the study is completed. For example: Deidentified research data will be made publicly available when the study is completed and published.

Thank you for the comment. We are using data that is already publicly available from published qualitative studies. Any data we generate will be reported in the findings section of the results manuscript and any additional data will be provided as appendices and on PROSPERO following study completion. We have stated the following in the protocol:

“We will report this study in accordance with the Enhancing Transparency in Reporting the synthesis of Qualitative research (ENTREQ) statement [69]. We will publish our findings in a peer-reviewed journal, within 12 months of completing the study and further disseminate via relevant clinical and academic conferences, public and patient advocacy groups, and social media. We aim to make our data FAIR, i.e., findable, accessible, interoperable, and reusable [70]. Finally, data generated as a result of the synthesis steps as outlined above, such as generated codes and themes, will be made available as appendices and on the PROSPERO database registration for this study.”

Introduction

2. Lines 121-122 – change “substantially” to “substantial” or “significant”? (I’m left wondering whether it remained significant).

Thank you. This has been amended in the text.

3. Can you please provide citations for the statement on line 132: “People living with obesity and pain have a higher pain incidence, severity and morbidity.”

Thank you for the comment. This statement in the text is a summary of the previous two paragraphs. Supporting citations have now been added.

AIM

1. How are the statements on lines 150-152 and line 167-168, different? The aim should probably use the phrase “Perspectives of PwO living with CP.” Prior to reading the Aim, the Introduction should definitely elaborate on the distinction between “synthesized lived experiences” and “perspectives of….” It is not clear from the Introduction, so the reader arrives at the Aim without a solid understanding of how this study is different from others, and what contribution it would make to the literature. Based on lines 150-152 of the Introduction, the Aim could be strengthened by saying something about how the data will be used to “understand… the complex interaction between the two.

**It is critical that the authors distinguish between the published reviews of the “lived experience” and what their study will provide/how their study is different. That would help to justify why your study is needed. I recommend using the Introduction to help make your case. To elaborate just a bit further, I read on line 198 that you’ll include studies with “first person experiences…” Can you clearly distinguish between your data vs data from already published syntheses of lived experiences?

Thank you for your time and these comments, which we have found very helpful.

Our view is that the perspectives of people with obesity are based on their personal lived experiences. As this QES will be synthesising qualitative studies conducted with people with firsthand experiences, in this specific context of a QES these concepts (‘perspectives’ versus ‘experiences’) are somewhat synonymous as there is no third party ‘objective’ evaluation of the veracity people’s experiences with both obesity and chronic pain (due to the prespecified methodology inclusion criteria of the studies) – only the first-person subjective recalling and reporting. Additionally, as each study findings are the interpretations of the study authors, rather than the direct transcripts from participant interviews, the concept of ‘experiences’ feels more holistic and accurate to describe impression of the lives of participants.

This has been a conscious linguistic choice from the authors in order to make the protocol more accessible and less repetitive. We have however, tried to amend where possible to assist with clarity as per your suggestions, which we appreciate greatly. Please see the tracked changes version for amendments highlighted in red. 

With reference to previous studies conducted, please see our comments below.

The search strategy for the study Cooper et al. 2017 only included published papers up to 23 July 2016 and their findings are based on four studies, three of which are more than a decade old. While these findings are valuable, there have been several further developments both in in the body of scientific work and public understanding regarding obesity and pain since the time of the original studies.

Additionally, Cooper et al. 2017 included both overweight and obesity, whereas our inclusion criteria is limited to only those with obesity to reflect an understanding that these populations may not have similar experiences of pain (such as the amplification of pain due to weight-related stigma from living in a larger body). We anticipate that there could be some crossover of included studies that meet our inclusion criteria. However, as there has been increased focus on PwO in more recent years, we anticipate the additional inclusion of studies conducted more recently and those only specific to people living with obesity and CMP.

We have included the following new text in the main text of the protocol to assist with clarity in this matter:

“While a valuable addition to the body of knowledge, the scope of this study [Farrell, et al. 2021] was very broad and failed to provide any explicit insights into the experiences of those specifically living with both obesity and chronic musculoskeletal pain, as this was not their focus.”

“While Cooper et al. 2017 has also provided great insights into the experiences of people overweight and those living with multimorbidity of obesity and CMP, three of the four studies included are now more than a decade old, with several important advancements in the understanding of both pain and obesity in that time.”

2. How will the anthropometric data and pain characteristics data be used to inform your aim?

The anthropometric data will be used in conjunction with the data extracted from the included papers to inform, support or exclude any themes generated (as an example, themes associated specifically with gender or age).

---

## [Editor Report · Decision Letter 2]

27 Mar 2024

The Experiences of People Living with Obesity and Chronic Pain: A Qualitative Evidence Synthesis (QES) Protocol

PONE-D-23-17779R2

Dear Dr. Hinwood,

We’re pleased to inform you that your manuscript has been judged scientifically suitable for publication and will be formally accepted for publication once it meets all outstanding technical requirements.

Kind regards,

Zeina El Ali, Ph.D.

Guest Editor

PLOS ONE